

# Competing topological orders in three dimensions: X-cube versus toric code

M. Mühlhauser[1], K. P. Schmidt[1], J. Vidal [2] and M. R. Walther[1*]

**1** Department Physik, FAU Erlangen-Nürnberg, Germany
**2** Sorbonne Université, CNRS, Laboratoire de Physique Théorique de la Matière Condensée, LPTMC, F-75005 Paris, France

* matthias.walther@fau.de

## Abstract

We study the competition between two different topological orders in three dimensions by considering the X-cube model and the three-dimensional toric code. The corresponding Hamiltonian can be decomposed into two commuting parts, one of which displays a self-dual spectrum. To determine the phase diagram, we compute the high-order series expansions of the ground-state energy in all limiting cases. Apart from the topological order related to the toric code and the fractonic order related to the X-cube model, we found two new phases which are adiabatically connected to classical limits with nontrivial sub-extensive degeneracies. All phase transitions are found to be first order.



# 1 Introduction

Quantum systems with topological order are an important research field due to their intriguing physical properties as well as their potential relevance for quantum technological applications. In two dimensions, these systems are essentially characterized by long-range entanglement and exotic excitations called anyons [1, 2], which have quantum exchange statistics distinct from bosons and fermions (see Ref. [3] for a review). These anyonic particles are at the heart of topological quantum computing [4, 5] and have been observed unambiguously in quantum Hall systems only very recently [6, 7]. Other examples of physical systems where topologically-ordered states play an important role are frustrated quantum magnets and synthetic matter in quantum-optical platforms [8–15].

During these last years, topological order in three dimensions (3D) gained a lot of interest. Some properties such as a topology-dependent ground-state degeneracy are very similar to two-dimensional topological order. Furthermore, in three-dimensional topologically-ordered systems point-like anyonic excitations are excluded but nontrivial statistics can be found for extended objects such as membranes. However, in 3D, one must distinguish between two main categories of topologically-ordered long-range entangled ground states [16, 17] depending on whether their degeneracy is finite [18–21] or sub-extensive with the system size [22–35]. The topological order for systems with sub-extensive ground-state degeneracy is called fracton order. One defining characteristic of fracton phases is that their elementary excitations have a restricted mobility under the action of local operators so that they are considered as attractive candidates for 3D quantum memories [26].

Paradigmatic examples of these two categories of topological order are the 3D toric code (TC) model [19, 20], which is a direct extension of the celebrated model introduced by Kitaev in two dimensions [4] for fault-tolerant quantum computation and has a finite ground-state degeneracy, and the X-cube model (XC) proposed by Vijay, Haah, and Fu [28], which has a sub-extensive ground-state degeneracy.

In the absence of any local order parameter, the study of transitions between topological quantum phases of matter is a challenging problem. In two dimensions, the concept of anyon condensation [36] and topological symmetry breaking provides a general framework [37] to understand some of these transitions [38–40]. Other salient examples are the Kitaev's honeycomb model which exhibits a transition between an achiral topological phase and a chiral topological phase [41], the string-net model [42] that allows to investigate the competition between different topological achiral phases obeying the same fusion rules [43–45], or multi-layer systems [35, 46]. To our knowledge, similar studies are still missing in 3D.

The goal of the present work is to investigate the competition between two different types of topological orders by considering the interplay between the TC and the XC phases on the cubic lattice. As shown below, the corresponding Hamiltonian can be split into two commuting parts that are analyzed separately. Interestingly, one of these parts has a self-dual spectrum. We determine the phase diagram in the full four-dimensional parameter space using this self-duality as well as high-order series expansions of the ground-state energy in all limiting cases. Apart from the TC and XC phases, the phase diagram displays two additional phases, dubbed X- and Z-phases, that are connected to classical limits with sub-extensive degeneracies. All phase transitions are found to be first order.

The paper is organized as follows: in Sec. 2 we introduce the model and we recall the main properties of the limiting cases. Then, we show that one can recast the full Hamiltonian in two sets of commuting operators allowing for a simpler analysis of the phase diagram which is discussed in Sec. 3. We conclude our findings and give some perspectives in Sec. 4.

## 2 Model

We consider the interplay between the TC and the XC. Microscopic degrees of freedom are spins-1/2 located on the links of the cubic lattice (see Fig. 1). The corresponding Hamiltonian is given by

$$\mathcal{H} = \mathcal{H}_{\text{TC}} + \mathcal{H}_{\text{XC}}, \tag{1}$$

with

$$\mathcal{H}_{\text{TC}} = -J_* \sum_* X^* - J_\square \sum_\square Z^\square, \tag{2}$$

$$\mathcal{H}_{\text{XC}} = -J_+ \sum_+ X^+ - J_{\oplus} \sum_{\oplus} Z^{\oplus}, \tag{3}$$

where $\square$ and $\oplus$ represent elementary faces and cubes of the cubic lattice, whereas $*$ and $+$ label the two different vertex operators of the TC and XC, respectively (see Fig. 1 for illustration). Without loss of generality, we assume non-negative couplings $J_*$, $J_\square$, $J_+$, and $J_{\oplus}$ for the rest of the paper.

Denoting by $\sigma_i^\alpha$ the usual Pauli matrices with $\alpha = x, y, z$ acting on the link $i$, the operators of the TC are defined as

$$X^* = \prod_{i \in *} \sigma_i^x, \tag{4}$$

$$Z^\square = \prod_{i \in \square} \sigma_i^z, \tag{5}$$

where the products run over the six spins (four spins) of $*$ ($\square$). The XC operators are

$$X^+ = \prod_{i \in +} \sigma_i^x, \tag{6}$$

$$Z^{\oplus} = \prod_{i \in \oplus} \sigma_i^z, \tag{7}$$

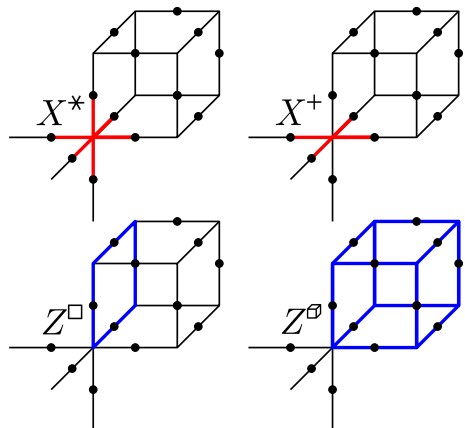

Figure 1: Illustration of the four operators on the cubic lattice which are contained in $\mathcal{H}$. *Left*: Operators $X^*$ and $Z^\square$ of the TC. *Right*: Operators $X^+$ and $Z^{\oplus}$ of the XC. Note that we show only one of three possible orientations of the $X^+$ and $Z^\square$ operators. Red (blue) operators are multi-spin interactions built by $\sigma_i^x$ ($\sigma_i^z$) Pauli matrices. Here $i$ denotes the links of the cubic lattice and the spin-1/2 degrees of freedom are indicated by filled black circles.

where the products run over the four spins (twelve spins) of $+$ ($\boxplus$). The four operators (4)-(7) have eigenvalues $\pm 1$. We stress that all pairs of operators commute unless they have different spin flavors and share an odd number of spins. Hence, the Hamiltonian (1) is not exactly solvable for arbitrary couplings but there are some limiting cases where $\mathcal{H}$ can be solved analytically.

## 2.1 Limiting cases

These limits are connected to the four different phases present in the phase diagram discussed in the next section.

*TC phase:* For $J_+ = J_{\boxplus} = 0$, the system reduces to the pure TC which is exactly solvable [19–21] since $[X^\star, Z^\square] = 0$ for all $\star$ and $\square$. The TC has a finite ground-state degeneracy which only depends on the genus of the 3D surface [16,18], e.g., the degeneracy is $2^3$ on a 3-torus. These ground states have a finite topological entropy and can be seen as 3D generalizations of the loop gas ground state of the conventional 2D toric code [19–21]. Furthermore, gapped elementary excitations correspond to point and spatially extended particles displaying a semionic mutual statistics. In the following, we call the phase adiabatically connected to the limit where $J_+ = J_{\boxplus} = 0$, the TC-phase.

*XC phase:* For $J_\star = J_\square = 0$, the system corresponds to the exactly solvable XC [28]. Its ground-state degeneracy depends not only on the topology but also on the geometry of the system [30,47]. For instance, on a 3-torus with linear extensions $L_1$, $L_2$, and $L_3$, the ground-state degeneracy equals $2^{2(L_1+L_2+L_3)-3}$ [17,30]. The ground states of the XC have a topological entanglement entropy which scales sub-extensively with the linear system size [48,49] and may again be seen as a generalized loop gas [50]. The XC is known to display type-I fracton topological order [28], i.e., its elementary excitations are either immobile or have a dispersion with dimensional reduction upon the action of local operators. The immobile fracton excitation of the XC corresponds to a single cube excitation with eigenvalue $-1$ of one $Z^{\boxplus}$. The other elementary excitations of the XC are one-dimensional particles related to pairs of $X^+$ operators with eigenvalue $-1$ at the same vertex. We call the phase adiabatically connected to the limit where $J_\star = J_\square = 0$, the XC-phase.

*Classical limits:* In the limit where $J_{\boxplus} = J_\square = 0$ ($J_+ = J_\star = 0$), the only operators in the Hamiltonian are products of $\sigma_i^x$ ($\sigma_i^z$). Eigenstates of $\mathcal{H}$ are thus trivial product states and the ground states are all states with eigenvalues $+1$ for these operators. For periodic boundary conditions (3-torus with linear extension $L$), one finds a non-trivial ground-state degeneracy $2^{3L^2}$ for $J_{\boxplus} = J_\square = 0$ and $2^{3(L^2-L-1)}$ for $J_+ = J_\star = 0$. For $J_{\boxplus} = J_\square = 0$, the different ground states are distinguished by a set of non-local commuting operators defined as product of $\sigma_i^z$ on straight lines. For $J_+ = J_\star = 0$, ground states are distinguished by a set of non-local commuting operators defined on non-contractible tubes and planar membranes (details about these operators are given in Appendix A). As a direct consequence, the ground-state degeneracy of these phases is expected to be robust with respect to small perturbations in the couplings $J_{\boxplus}, J_\square$ or $J_+, J_\star$. In the following, phases connected to the two classical limits will be called X- and Z-phase for obvious reasons.

## 2.2 Exact decomposition and self-duality

The essential ingredient to determine the ground-state phase diagram in the full parameter space relies on another decomposition of the Hamiltonian. Instead of writing $\mathcal{H} = \mathcal{H}_{\text{TC}} + \mathcal{H}_{\text{XC}}$ with $[\mathcal{H}_{\text{TC}}, \mathcal{H}_{\text{XC}}] \neq 0$, one can recast it as $\mathcal{H} = \mathcal{H}_A + \mathcal{H}_B$ with

$$\mathcal{H}_A = -J_* \sum_* X^* - J_\boxdot \sum_\boxdot Z^\boxdot, \tag{8}$$

$$\mathcal{H}_B = -J_+ \sum_+ X^+ - J_\Box \sum_\Box Z^\Box, \tag{9}$$

and, as can be easily checked, $[\mathcal{H}_A, \mathcal{H}_B] = 0$. Interestingly, the spectrum of $\mathcal{H}_A$ is exactly self-dual (up to degeneracies) so that the phase diagram of $\mathcal{H}_A$ must be symmetric with respect to the self-dual point $J_* = J_\boxdot$. To prove this self-duality, let us introduce the following pseudospin-1/2 operators $\tau_\nu^z = X^*$ defined on the vertices of the original cubic lattice $\Lambda$. Then, it is easy to see that $Z^\boxdot$ acts like $\prod_{\nu \in \boxdot} \tau_\nu^x$ so that $\mathcal{H}_A$ becomes

$$\mathcal{H}_A^\tau = -J_* \sum_{\nu \in \Lambda} \tau_\nu^z - J_\boxdot \sum_{\boxdot \in \Lambda} \prod_{\nu \in \boxdot} \tau_\nu^x, \tag{10}$$

which describes a transverse-field spin-1/2 model with eight-spin interactions on a cubic lattice.

Similarly, if one introduces $\widetilde{\tau}_\nu^z = Z^\boxdot$ defined on vertices of the cubic lattice $\widetilde{\Lambda}$ spanned by the center of each elementary cube of $\Lambda$, the operator $X^*$ acts like $\prod_{\nu \in \boxdot} \widetilde{\tau}_\nu^x$ so that $\mathcal{H}_A$ becomes

$$\mathcal{H}_A^{\widetilde{\tau}} = -J_* \sum_{\nu \in \widetilde{\Lambda}} \prod_{\nu \in \boxdot} \widetilde{\tau}_\nu^x - J_\boxdot \sum_{\nu \in \widetilde{\Lambda}} \widetilde{\tau}_\nu^z. \tag{11}$$

Since $\Lambda$ and $\widetilde{\Lambda}$ are both cubic lattices, the spectrum of $\mathcal{H}_A^\tau$ and the one from $\mathcal{H}_A^{\widetilde{\tau}}$ are obtained from the other by exchanging $J_* \leftrightarrow J_\boxdot$. As a consequence, the spectrum of $\mathcal{H}_A$ is invariant under this exchange and, hence, self-dual. Of course, the mapping described above does not preserve the degeneracies of the spectrum. Thus, the self-duality of $\mathcal{H}_A$ is only exact, up to degeneracies. For a very similar discussion in two dimensions, see Refs. [51, 52].

In contrast, $\mathcal{H}_B$ is not self-dual as can be directly seen in the series expansions of the ground-state energy given in Appendix B.

# 3 Phase diagram

The decomposition of $\mathcal{H}$ into two commuting parts ($[\mathcal{H}_A, \mathcal{H}_B] = 0$) allows one to build the full phase diagram from the ones of $\mathcal{H}_A$ and $\mathcal{H}_B$, separately. We emphasize that this decoupling implies that the phase diagram only depends on the two ratios $J_\boxdot/J_*$ and $J_+/J_\Box$ driving the transition of $\mathcal{H}_A$ and $\mathcal{H}_B$, respectively.

The self-duality of $\mathcal{H}_A$ implies that if there is only one transition point, it can only occur at the self-dual point where $J_\boxdot/J_* = 1 \equiv \eta_A$. The ground-state energy of $\mathcal{H}_A$ computed perturbatively in the limit where one of the coupling dominates is displayed in Fig. 2 (see Appendix B for analytical expressions).

Similarly, assuming the existence of a unique transition point in the phase diagram of $\mathcal{H}_B$, we determined its position by extrapolating the crossing point of high-order series expansions for the ground-state energy in both limiting cases $J_+ \ll J_\Box$ and $J_\Box \ll J_+$ (see Appendix B). We found a transition point at $J_+/J_\Box \simeq 1.012 \equiv \eta_B$ (see Fig. 2).

We stress that $\eta_A$ and $\eta_B$ are associated to first-order transitions. Hence, assuming a unique transition point for $\mathcal{H}_A$ and $\mathcal{H}_B$, we obtain the complete phase diagram of $\mathcal{H}$ shown in Fig. 3 which contains four distinct phases separated by first-order transition lines. For small $J_+/J_\Box$ and $J_\boxdot/J_*$ the system is in the TC-phase. For large $J_+/J_\Box$ and $J_\boxdot/J_*$, one gets the fractonic XC-phase. In the limit $J_+/J_\Box \gg 1$ and $J_\boxdot/J_* \ll 1$, one finds a phase essentially driven by the

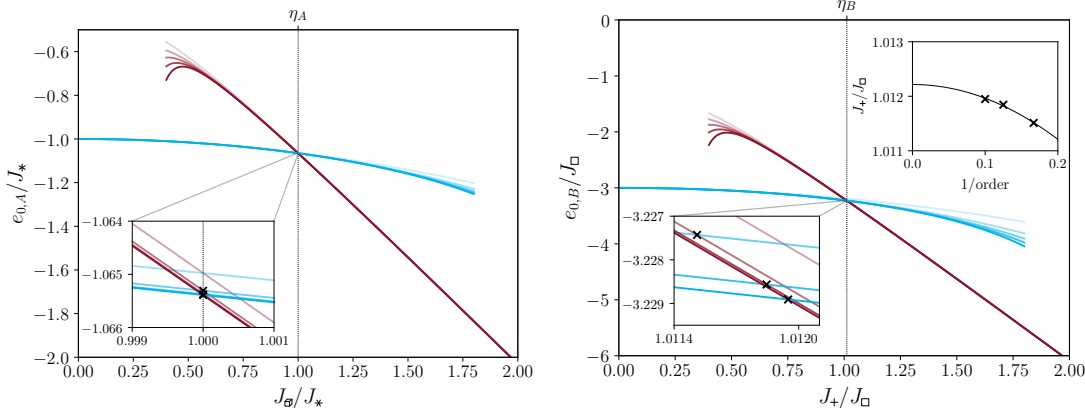

Figure 2: Ground-state energy per vertex of $\mathcal{H}_A$ (left panel) and $\mathcal{H}_B$ (right panel) obtained from high-order series expansions. Bare series of even orders from 2 to 10 are shown as solid lines from light to dark colors. Vertical dotted lines indicate the phase transition points $\eta_A$ and $\eta_B$. *Lower insets*: zoom of the ground-state energy close to the phase transition points. Crosses indicate the intersection points of the series expansions at order 6, 8, and 10. *Upper inset*: Crossing points as a function of the inverse order. The thin solid line serves as a guide to the eyes.

two operators consisting of $\sigma_i^x$, i.e., the X-phase. Similarly, when $J_+/J_\square \ll 1$ and $J_\boxplus/J_* \ll 1$, the phase is mainly determined by the two operators consisting of $\sigma_i^z$, namely, the Z-phase.

Interestingly, we emphasize that a direct transition between the XC- and the TC-phase requires a fine-tuning of the parameters.

# 4 Conclusions

In this work we investigated the competition between the two most paradigmatic representatives of 3D topological order. An exact decomposition of the system allows for a quantitative determination of the ground-state phase diagram in the full parameter space. Apart from the TC- and XC-phase, the phase diagram contains two additional phases, the X- and Z-phases, that are connected to limiting cases where operators with either $\sigma_i^x$ or $\sigma_i^z$ Pauli matrices dominate. In the purely classical limits, we find non-trivial sub-extensive ground-state degeneracies which are robust perturbatively. However, a better understanding of the quantum nature of the X- and Z-phases would be valuable but it is beyond the scope of the present work.

In the derivation of the full phase diagram we assumed that $\mathcal{H}_A$ and $\mathcal{H}_B$ display a single phase transition which is found to be first order. Unfortunately, the existence of intermediate phases can not be ruled out by our approach. Although we consider unlikely the existence of intermediate phases, an unbiased numerical investigation would be valuable.

KPS acknowledges financial support by the German Science Foundation (DFG) through the grant SCHM 2511/11-1.

# A  Ground-state degeneracy of X- and Z-phases

In this appendix we compute the ground-state degeneracy (GSD) of the X- and Z-phase in the classical limit on a 3-torus.

In order to access the GSD of the X-phase and the Z-phase in the classical limit on a 3-

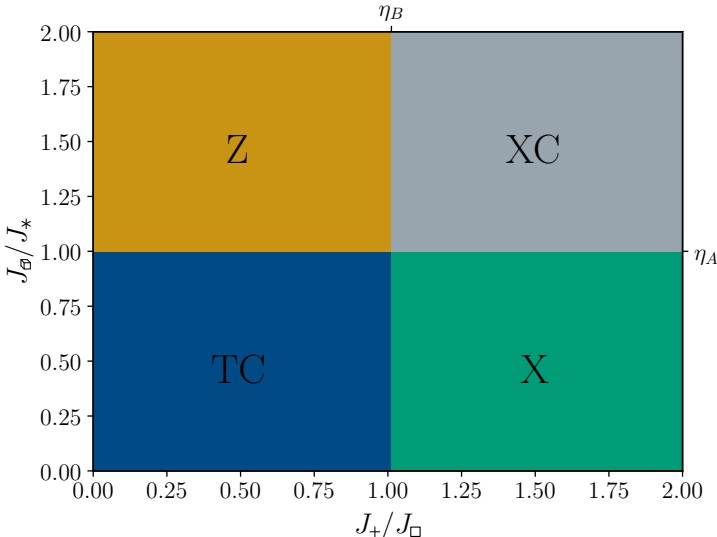

Figure 3: Ground-state phase diagram of $\mathcal{H}$ as a function of $J_+/J_\square$ and $J_\boxminus/J_*$. Vertical and horizontal lines correspond to $\eta_A$ and $\eta_B$ (see text). Topological TC- and XC-phase are displayed in blue and gray, whereas the X- and Z-phases are shown in green and brown, respectively.

torus of dimensions $L_1 \times L_2 \times L_3$ we describe the operators whose eigenvalues distinguish between the different ground-states. Importantly, these operators are non local. As shown below, we found that $\log_2 \mathrm{GSD}$ obeys an area law, contrasting with the linear behaviour of the XC-phase [17,30] and the constant value of the TC-phase [19–21].

## A.1 X-phase

In the X-phase the relevant operators are products of $\sigma_i^z$ operators acting on straight lines and forming non-contractible loops (see Fig. 4 for illustration). Since all these operators are independent and mutually commute, there are

$$\log_2 \mathrm{GSD} = L_1 L_2 + L_2 L_3 + L_3 L_1, \tag{12}$$

loops on the 3-torus.

We checked these expressions numerically, on (small) finite-size systems.

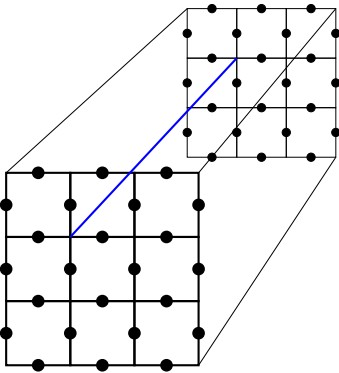

Figure 4: Illustration of a non-local "line"-operator corresponding to the product of $\sigma_i^z$ along the blue line assuming periodic boundary conditions.

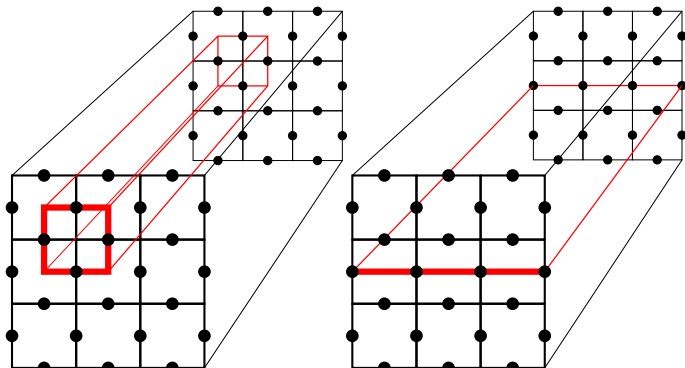

Figure 5: Illustration of non-local tube and plane operators assuming periodic boundary conditions. Left: Tube operators, which correspond to the action of $X^\star$ along a straight line. Right: Plane operators, which correspond to the action of $\sigma_i^x$ on all sites in the indicated plane.

## A.2 Z-phase

In the Z-phase there are two types of operators whose eigenvalues distinguish between the different ground-states, planes of $\sigma_i^x$ which are precisely halfway in between the lattice planes and non-contractible tubes of $\sigma_i^x$ which correspond exactly to the action of $X^\star$-operators along a straight line. Two plane-operators correspond exactly to the product of the tube operators inside a layer, so not all tube operators inside a layer are independent. These non-local tube and plane operators are illustrated in Fig. 5. In total, we have $\sum_i L_i$ "plane" operators and $\sum_{i<j}(L_i-1)(L_j-1)$ "tube" operators. Accordingly, one gets

$$\log_2 \mathrm{GSD} = \sum_i L_i + \sum_{i<j}(L_i-1)(L_j-1). \tag{13}$$

## B    Series expansions

In this Appendix, we give high-order series for the ground-state energy per vertex and briefly comment on the methods to derive them.

We computed several high-order series expansions using the Löwdin method [53]. This method has been applied successfully in related contexts like the robustness of effective cluster states in measurement-based quantum computation [54] as well as topological string-net phases [55]. Furthermore, the application of the Löwdin method to perturbed topological models is well described in Ref. [56] and we therefore focus on the central aspects for the current problem. Here, we perform the series expansion separately for $\mathcal{H}_A$ and $\mathcal{H}_B$ and calculate the ground-state energy for all perturbative limits of $\mathcal{H}_A$ and $\mathcal{H}_B$. The series expansions of the ground-state energy of the original Hamiltonian (1) are then simply given by the sum of the ground-state energies of $\mathcal{H}_A$ and $\mathcal{H}_B$ in the appropriate limits. We stress that the obtained series for the ground-state energy are valid for the entire ground-state manifold in all considered limits. This is a direct consequence of the fact that ground states of the same manifold are only connected by non-local operators so that the degeneracy remains intact at any finite order of perturbation theory. Accordingly, the calculation of the ground-state energy up to any finite perturbation order can be performed on any state in the ground-state manifold without loss of generality.

The calculation is most efficiently done via a full graph decomposition using a linked-cluster expansion. Since $\mathcal{H}_A$ and $\mathcal{H}_B$ only contain multi-spin interactions linking multiple degrees of freedom, a natural formulation of the linked-cluster expansion is therefore performed in terms of hypergraphs [57]. A hypergraph is a generalization of a graph where edges can link more than two vertices. Technically, we generate all linked subclusters up to a given size [58, 59] and sort them into isomorphism classes of hypergraphs using their König representation [60, 61]. During this procedure non-contributing subclusters are discarded as early as possible using heuristics adapted from Refs. [62, 63]. This allows us to determine the ground-state energy per vertex for $\mathcal{H}_A$ and $\mathcal{H}_B$ in the four perturbative limits up to order 10. The corresponding series for the full problem $\mathcal{H}_A + \mathcal{H}_B$ can then be straightforwardly extracted.

For the ground-state energy of $\mathcal{H}_A$ per vertex up to order ten, we find

$$
\begin{aligned}
e_{0,A}^{J_* \ll J_\oplus} = & -J_\oplus - \frac{J_*^2/J_\oplus}{16} - \frac{71 J_*^4/J_\oplus^3}{28672} \\
& - \frac{5357137 J_*^6/J_\oplus^5}{16184770560} - \frac{15573579216301097 J_*^8/J_\oplus^7}{235160106814144512000} \\
& - \frac{23772819421675595994611334959 J_*^{10}/J_\oplus^9}{1465811223338361510040279449600000}.
\end{aligned}
\tag{14}
$$

Because of the exact self-duality the series in the opposite limit $J_\oplus \ll J_*$ is easily obtained by exchanging $J_*$ and $J_\oplus$ in the above expression.

For the ground-state energy of $\mathcal{H}_B$ per vertex up to order ten we obtain

$$
\begin{aligned}
e_{0,B}^{J_\square \ll J_+} = & -3J_+ - \frac{3 J_\square^2/J_+}{16} - \frac{195 J_\square^4/J_+^3}{28672} \\
& - \frac{7052113 J_\square^6/J_+^5}{6936330240} - \frac{2392948067252749 J_\square^8/J_+^7}{10853543391422054400} \\
& - \frac{58797670263954034869471 5843 J_\square^{10}/J_+^9}{99715049206691259186413568 00000},
\end{aligned}
\tag{15}
$$

$$
\begin{aligned}
e_{0,B}^{J_+ \ll J_\square} = & -3J_\square - \frac{3 J_+^2/J_\square}{16} - \frac{3 J_+^3/J_\square^2}{128} \\
& - \frac{195 J_+^4/J_\square^3}{28672} - \frac{4455 J_+^5/J_\square^4}{1605632} \\
& - \frac{14445391 J_+^6/J_\square^5}{12138577920} - \frac{286541706167 J_+^7/J_\square^6}{489427461734400} \\
& - \frac{21431205246868721 J_+^8/J_\square^7}{70548032044243353600} \\
& - \frac{1685552044984622774 14651 J_+^9/J_\square^8}{1016907553098541396131840000} \\
& - \frac{4306666634113068997936331017 J_+^{10}/J_\square^9}{4580660072932379718875873 2800000}.
\end{aligned}
\tag{16}
$$

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
