# Peer review of "Competing topological orders in three dimensions"

_SciPost Physics, doi:SciPost Phys. 12, 069 (2022)_

## Round 1 · Referee Report · Helene Spring (Referee 1) · 2021-8-23

Report

The authors study a four-parameter Hamiltonian composed of the three-dimensional toric code Hamiltonian and the X-Cube Hamiltonian. The ground-state degeneracy and the high-order series expansion of the ground state energies are calculated in four different limits. The expressions for the energy are extended to other regions of the parameter space in order to define a phase diagram. The authors present two as of yet unreported phases arising from the competition between two known topologically ordered systems. Despite the fact that this field is not my principal area of expertise, the manuscript was accessible and clear to read. I recommend this manuscript for publication, once some structural errors in the text are corrected. These are as follows:

1) In Figure 2, the descriptions of the left and right insets appear to be switched.

2) The ‘square’ symbol in Z_square is not correctly rendered in the manuscript. (An alternative could be to simplify the notation to Z_f and Z_c, for ‘face’ and ‘cube’ operators respectively to increase legibility. )

3) In Appendix A, it seems as though the definitions of the X- and Z- phases are inverted with respect to their definitions in the main text.

I have one remark about the content of the manuscript that I would like to see addressed:

4) Appendix B lists the methods used to calculate the analytical results for four parameter limits. This section is quite concise, and lacks elaboration on how the Lowdin method is used to obtain the results. This method is not commonly used in papers on 2D or 3D systems with topological order; as such, either expanding this section, referring to specific sections of the original Lowdin paper or adding references that use this method in comparable systems would be helpful to the reader.

I also have some questions and comments for the authors, but I leave the decision of whether to use them to modify their manuscript.

5) The manuscript explores limits of a Hamiltonian composed of the 3D toric code Hamiltonian and the X-Cube Hamiltonian. The authors bring to light the existence of two additional phases of the Hamiltonian, X- and Z-phases, and their ground state degeneracies are studied. It might be more appropriate to include these results in the main text rather than relegate them to the Appendices. It is unclear to me why a more substantial discussion of these new phases is left for future work, as a clear motivation for an additional, self-contained study of these phases is not provided in the outlook section.

6) The phase diagram is constructed from four separate points in phase space, then the transition points are found by extending the analytical solutions of the energy at these points to other regions in phase space, in order to find where they intersect. Since the authors mention that there is no guarantee of a single transition point between phases, it seems premature to present a phase diagram. The assumption of a unique crossing point is a hypothesis that could be tested numerically. Therefore this task could be left to future numerical work.

---

## Round 1 · Referee Report · Anonymous (Referee 2) · 2021-9-26

Strengths

Nice investigation of the phase-diagram of a three-dimensional X-cube+toric-code model.

Weaknesses

At first sight, not very attractive for a reader.

Report

Following up on previous publications by the same group on related two- [53] and three-dimensional models [39,54], the present work investigates the competition between two different topological orders in a three-dimensional X-cube+toric-code model. The total Hamiltonian can be decomposed into two commuting parts, one of which displays a self-dual spectrum. The phase diagram is derived from high-order series expansions of the ground-state energy. The central result is the ground-state phase diagram Fig. 3 for the full model that exhibits four phases. Result are reliable in the limits where one of the coupling constants vanishes while it would be good to validate the absence of further phases by other methods.

Once one sits down and reads this manuscript, this is a nice short note. However, the authors might be underselling their work, e.g., for the following reasons:
1- The title of the manuscript is a bit unspecific.
2- The presentation of the new results is quite compact and overshadowed by a long review of previous results (until the end of page 4) and a reference list that amounts to about a third of the manuscript. In particular the broadside citation of Refs. [16-40] in the second paragraph of the Introduction is not very helpful for the reader.

Beyond the above, I have a few more specific comments that I list among the “Requested changes”. Once the authors have addressed these concerns, I expect the present manuscript to be a nice piece of work that is suitable for publication in SciPost Physics.

Requested changes

1- The title appears a bit too generic to me. I recommend to look for one that is more specific as far as model and/or method are concerned.
2- Purge unnecessary references among [16-40] or at least make the discussion in the second paragraph of the Introduction more specific.
3- The review copy has a number of white spaces (e.g., after "where" in the line below Eq. (3)) which leads me to suspect that a certain symbol (a square?) was systematically lost. Please check and correct as appropriate.
4- First line of caption of Fig. 2: there are only "left" and "right" panels, but no "upper" and "left" ones.
5- Figure 5 is not referenced; it appears to be related to appendix A.2, but this is not stated explicitly.
6- Some details underlying the series presented in Appendix B are unclear. In particular, since ground states are supposed to be degenerate, it is not clear if these series are valid for any of them, or if they have computed with the help of a specific choice of ground state.
7- Make sure that the names "Hall", "Moore-Read", and "Ising" start with an upper-case letter in the titles of Refs. [45,46], [47], and [53], respectively.
8- The first and second author of Ref. [50] should be separated by a comma and not by "and".
9- The first word of the title of Ref. [66] seems to be misspelled.

---

## Round 2 · Referee Report · Helene Spring (Referee 1) · 2021-10-15

Report

I thank the authors for considering my questions and suggestions, which they have addressed in both their response and in the new version of the manuscript. Following the modifications they have implemented, I am happy to recommend the publication of the manuscript in its present form.
  • validity: -
  • significance: -
  • originality: -
  • clarity: -
  • formatting: -
  • grammar: -

Author:  Kai Phillip Schmidt  on 2021-10-28  [id 1886]

(in reply to Report 1 by Helene Spring on 2021-10-15)

We thank the referee for her positive feedback and the recommendation for publication of our article in SciPost.

---

## Round 2 · Referee Report · Anonymous (Referee 2) · 2021-10-26

Report

With their revised version, the authors have addressed most previous concerns. Apart from some possible typographic errors that I list as "Requested changes", there are just some details that have not really been addressed: 1- The question about series expansion for degenerate ground states (item 6 of previous Report 2) has not really been addressed. Ok, there is a statement now on page 9 that the series are valid for the entire ground-state manifold, but this does still not explain how the computation has actually been carried out. 2- In their response to item 4 of Report 1, the authors claim to have added a link to a PhD thesis. I understand that this is Ref. [56], but I was not able to find a link.

I believe that the two points above and the suggestions for proofreading are minor items that can be taken care of during the production/proof stage. Otherwise, this is now really a nice short note. Consequently, I recommend publication of the manuscript in SciPost Physics.

Requested changes

Requests: 1- Add a sentence to Appendix B to explain how the degenerate ground states are treated during the calculation of the series. 2- Make sure that DOIs/links are available for all references, in particular Ref. [56].

A few further suggestions for proofreading: 3- Line 4 of abstract: "displays" rather than "displaying"? 4- Lines 17,18 of Introduction: "The topological order ... ARE called fracton order" -> "The topological order ... IS called fracton order" ? 5- Line 5 of section 2.1: remove space in "e.g. ,". 6- Line 4 of caption of Fig. 1: "build" -> "built". 7- Line 11 of paragraph "XC phase" on page 4: "-1" -> "$-1$". 8- Line 7 of section 3: no full stop between "Fig. 2" and "(see". 9- Last sentence of caption of Fig. 2: Insert "The" before "thin solid"? 10- Page 6, 4 lines below Fig. 2: Insert "the" before "TC-phase"? 11- Line 4 of page 9: I believe that after "consequence", the correct preposition would be "of" rather than "from".

  • validity: high
  • significance: good
  • originality: good
  • clarity: good
  • formatting: excellent
  • grammar: excellent

Author:  Kai Phillip Schmidt  on 2021-10-28  [id 1887]

(in reply to Report 2 on 2021-10-26)

We thank the referee for the feedback and the recommendation for publication of our article in SciPost apart from two requests and additional minor points. We have adressed the points as follows:

1-We added a sentence which should explain how the series is computed. 2-We made DOIs available where we could find them (For Refs [56] and [62] but not for Refs. [57],[58] and [63]). 3 to 11-We adjusted the respective parts according to the suggestions of the referee.

List of Changes

For the sake of clarity, all changes in the revised version are highlighted in red. The changes are essentially what has been requested by Referee 2.

---

## Round 2 · Author Response

Reply to Referee 1

We thank Referee 1 for her report and for considering our manuscript as suitable for publication. We provide thereafter a detailed answer to various comments made by Referee 1.

1- About the description of Figure 2: The problem comes from the layout of the two figures. With the actual presentation, the upper (lower) panel mentioned the caption must be replaced by left (right) panel. However, in each figure, the left inset located in the leftmost lower corner indeed represents a zoom of the ground-state energy near the transition point. We have now changed the caption to avoid any ambiguity in this description. We also added crosses in the lower inset of the left panel.

2- It seems that a figure containing the square plaquette was missing in the submission process to compile properly the manuscript. We added it and it should now look much better to the reader.

3- As explained in the text, the X-phase (Z-phase) is adiabatically connected to the point where only $\sigma_x$ ($\sigma_z$) Pauli matrices are present in the Hamiltonian. In the appendix A, the definition is exactly the same but the confusion of the referee may come from the fact that what we discuss the operators that commute with the Hamiltonian. In the X-phase (Z-phase), these operators are products of $\sigma_z$ ($\sigma_x$ ). Hence, everything is correct and consistent with the definitions given in the main text.

4- The Loewdin partition technique has been already applied at several instances in related problems by us. We therefore have added a sentence in the appendix B refering to these works and additionally giving a link to a phd thesis where more details on the actual application of this technique are described.

5- In the main text, we give the ground-state degeneracy of the X- and Z- phases and we believe that their derivation based on an exact counting of independent conserved quantities is rather suited for appendices. What is left for future work is the nature of these phases away from the extreme points where we can compute this degeneracy exactly. For instance, when $J_+ \neq 0$ we do not know what happens in the X-phase. Do we have a finite topological entropy ? What is the nature and the spectrum of the excitations ? These are typical questions that we would like to address but that are beyond the scope of the present work and that requires alternative approaches

6- The Hamiltonian (1) is not exactly solvable for arbitrary coupling. Thus, we use various approaches to understand its phase diagram. What we claim is that, if there is a unique quantum phase transition, our phase diagram is correct. As stated in the conclusion, this issue requires further (numerical) investigations, but, in any case, no exact answer can be given. Even numerically, it would be hard to decide the uniqueness of a transition point in the thermodynamical limit. Thus, we understand that our phase diagram can be considered as "likely" or "putative" but we clearly explain, in the text, under which hypothesis it is obtained.

Reply to Referee 2

We thank Referee 2 for her/his report and for considering our manuscript as suitable for publication. We provide thereafter a detailed answer to the points raised by Referee 2.

1- We extended the title to "Competing topological orders in three dimensions: X-Cube versus toric code".

2- We have rearranged and reduced the number of citations in this part of the introduction.

3- It seems that a figure containing the square plaquette was missing in the submission process to compile properly the manuscript. We added it and it should now look much better to the reader.

4- The problem comes from the layout of the two figures. With the actual presentation, the upper (lower) panel mentioned the caption must be replaced by left (right) panel. However, in each figure, the left inset located in the leftmost lower corner indeed represents a zoom of the ground-state energy near the transition point. We have now changed the caption to avoid any ambiguity in this description. We also added crosses in the lower inset of the left panel.

5- We have included a sentence within appendix A.2 referring to figure 5.

6- We added two sentences in appendix B.

7- We included the upper-case letters in the mentioned titles

8- We thank the referee for this comment and we did as suggested.

9- We thank the referee for this comment and corrected the title.

---

## Round 2 · List of Changes

Warnings issued while processing user-supplied markup:

  • Inconsistency: Markdown and reStructuredText syntaxes are mixed. Markdown will be used.
    Add "#coerce:reST" or "#coerce:plain" as the first line of your text to force reStructuredText or no markup.
    You may also contact the helpdesk if the formatting is incorrect and you are unable to edit your text.

List of Changes

For the sake of clarity, all changes (if appropriate) in the revised version are highlighted in red.

  • We have included the figure containing the square plaquette which was missing in the previous SciPost version.
  • Figure 2: We have updated the caption of figure 2 as suggested by both referees and we have added crosses in the left lower inset.
  • Page 8, reference to Figure 5.
  • Page 8, two additional sentence -> degeneracies
  • Ref 45,46,47 and 5, added upper-case letters in the titles
  • Ref 50, separated author name by comma instead of "and"
  • Ref 66, corrected the title

---

## Editorial Decision

published